# Incidental Hyperferritinemia in Very Young Infants with Mild Symptoms of COVID-19 Disease

**DOI:** 10.3390/children10050874

**Published:** 2023-05-12

**Authors:** Yuka Shishido, Haruhiko Nakamura, Tomohiro Nakagawa, Shinsuke Kanou, Takeshi Ito, Shota Kuwana, Chiharu Ota

**Affiliations:** 1Department of Pediatrics, Japanese Red Cross Ishinomaki Hospital, Ishinomaki 9868522, Japan; ykssd.ishi@gmail.com (Y.S.);; 2Department of Neurology, Miyagi Children’s Hospital, Sendai 9893126, Japan; 3Department of Pediatrics, Tohoku University Hospital, Sendai 9808574, Japan; 4Department of Development and Environmental Medicine, Tohoku University Graduate School of Medicine, Sendai 9808575, Japan

**Keywords:** early infantile COVID-19, hyperferritinemia, multisystem inflammatory syndrome, Japan

## Abstract

Background: The number of children infected with novel coronavirus disease 2019 (COVID-19), caused by severe acute respiratory syndrome coronavirus 2 (SARS-CoV-2), has increased during the outbreak of the Omicron strain. Hyperferritinemia has been reported in severe cases of COVID-19, and in children or neonates with multisystem inflammatory syndrome (MIS). Hyperferritinemia is considered to be one of the signs of MIS, but thus far, there have been few summarized reports on it. We retrospectively analyzed four infants less than 3 months of age with SARS-CoV-2 infections treated in our institution during the outbreak of the Omicron strain. Results: most patients were in good condition, but hyperferritinemia was observed in all of four cases. Conclusions: Hyperferritinemia can be observed in infantile COVID-19 patients even with mild symptoms. It is necessary to carefully monitor their clinical course and monitor the patients.

## 1. Introduction

In December 2019, the novel coronavirus disease 2019 (COVID-19), caused by severe acute respiratory syndrome coronavirus 2 (SARS-CoV-2), was first reported in Wuhan, China. Early reports indicated that children may be less likely to be infected and may present with more minor symptoms following infection [1]. However, the emergence of more contagious strains has changed this situation. As the number of infected children has increased, so too has the number of severe cases, and COVID-19 is now considered a threat to children. During the Omicron epidemic in 2022, a large number of children were infected, and the number of patients requiring medical care, including hospitalization, increased [2,3]. Moreover, in May 2020, reports from Europe and the United States reported that some children with a history of COVID-19 presented with symptoms similar to those of Kawasaki disease, including prolonged fever, skin rash, lymphadenopathy, diarrhea, and elevated inflammatory biomarkers, including ferritin, CRP, and D-dimer [4,5,6,7]. This condition was named multisystem inflammatory syndrome in children (MIS-C) [8]. Although MIS-C has been recognized in pediatric cases, reports of early infant cases, so-called MIS in newborns (MIS-N), remain scarce [9,10]. While appropriate treatment is required, there are still no reports describing the general clinical course of early infants, including mild cases, and the laboratory tests that should be monitored.

We have previously experienced a case of early infancy in which the patient had a rapid exacerbation of symptoms at 4–5 sick days. The case was a fifteen-day-old boy. He was admitted to our hospital with fever and poor feeding. Blood tests on admission showed no obvious abnormal findings. His respiratory condition had not worsened, but he had persistent fever and worsening peripheral coldness. On the seventh day of admission, a chest X-ray and blood tests were performed. The X-ray showed evidence of pneumonia, and the blood tests showed results suggestive of hypercytokinemia. After experiencing this case in the beginning of the COVID-19 outbreak in Japan among small children, we performed a detailed inpatient follow-up and multiple blood test evaluations of COVID-19 infection in early infant cases within the first 3 months of life with upper respiratory tract and respiratory symptoms, even if minor, or with fever and decreased feeding and activity. Therefore, we were able to obtain data on the general clinical course of the early infant, including mild cases.

In the present study, we report four cases of COVID-19 in early infants who showed mild symptoms without any signs of MIS-C/MIS-N or respiratory distress, but with increased serum markers of MIS [8], including ferritin.

We collected the patients’ characteristics including age, body weight, gender, prenatal information (gestational age, birth weight, maternal vaccination during pregnancy, and feeding methods), symptoms (fever, cough, nasal discharge, and appetite loss), and laboratory data. Blood tests were performed on admission and again about 4–5 days after admission. The background and course of the patients and the results of the blood tests performed were reviewed in light of previous reports. Patient data were retrospectively collected via electronic records of Japanese Red Cross Ishinomaki Hospital. Written informed consent was obtained from the guardians. These four patients were admitted from January to May 2022. Permission has been obtained from the Ethics Committee of Ishinomaki Red Cross Hospital. (No. 2023-1).

## 2. Cases

Case 1: A thirty-eight-day-old girl was admitted to our department with SARS-CoV2 infection, nasal discharge, and cough. She was born at 41 weeks of gestation in vacuum extraction with a birth weight of 3348 g. She showed transient tachypnea for the first few days with oxygen therapy and was discharged without any further complications. Her mother did not receive the SARS-CoV2 vaccine during pregnancy. At the age of 30 days, her relatives, who later tested positive for SARS-CoV2, visited her family. Two days later, she presented with nasal discharge and a cough. The SARS-CoV2 PCR results were positive 4 days before admission. Due to the increasing nasal discharge, the patient could not sleep well. On the day of admission, her oxygen saturation (SpO2) showed 92–94%. Upon physical examination, the patient weighed 4488 g. She had a body temperature of 36.5 °C, SpO2 of 97% on room air, and heart rate of 130–140 bpm without any respiratory distress. No crackles or wheezing were heard on auscultation. Chest radiography revealed no sign of pneumonia, bronchiolitis, or heart failure. Even without the specific symptoms as above, blood tests revealed elevated levels of aspartate transaminase (AST), 146 IU/L; alanine transaminase (ALT), 87 IU/L; lactate dehydrogenase (LDH), 370 IU/L; and ferritin, 570 ng/mL (Table 1). The patient showed slight exacerbation of her general condition after admission. On day 10 after onset (the 4th day of admission), her general condition and blood tests improved: AST, 91 IU/L; ALT, 82 IU/L; LDH, 300 IU/L; and ferritin, 368 ng/mL. The patient was discharged on day 11, and the blood test results improved further at the outpatient clinic on day 20: AST, 68 IU/L; ALT, 53 IU/L; LDH, 313 IU/L; and ferritin, 286 ng/mL (Table 1).

Case 2: A fifty-four-day-old girl was admitted with SARS-CoV-2 infection complicated by fever, cough, and poor feeding. She was born at 39 weeks of gestation with a birth weight of 2250 g. Her mother did not receive the SARS-CoV2 vaccine during pregnancy. No abnormalities were noted at the 1-month follow-up visit. Two days before the onset, her father became infected with SARS-CoV-2. On the day of admission, her body temperature was 38.4 °C with poor feeding. On physical examination, she did not show any symptoms of respiratory distress, with an SpO2 of 98%. Blood tests, including inflammatory markers and urinalysis, showed no abnormal findings. On the second day of hospitalization (the second day of onset), her fever resolved, and her general condition improved. On day 5, blood tests showed a mild elevation of aspartate aminotransferase (AST), 36 IU/L; lactate dehydrogenase, 299 IU/L; and ferritin, 319 ng/mL. On day 7, blood tests showed a slight aggravation of AST (42 IU/L), LDH (359 IU/L), ferritin (332 ng/mL), D-dimer (1.2 ng/mL), and CRP (1.1 mg/mL) (Table 1). We had previously encountered a COVID-19 case of a 14-day-old-boy who showed mild symptoms at the beginning and progressive pneumonia on day 7 found with thoracic computed tomography (CT). Thus, we examined the thoracic CT in this case, concerned about the exacerbation of the lung condition because of the slight change in the lab findings. CT revealed no findings associated with COVID-19 pneumonia. On day 10, her general condition was good with improved blood tests, including AST (33 IU/L), LDH (339 IU/L), and ferritin (306 ng/mL) (Table 1), and she was subsequently discharged from the hospital.

Case 3: A forty-seven-day-old girl was admitted to our hospital with SARS-CoV-2 infection and fever. She was born at 37 weeks of gestation with a birth weight of 2600 g. Her mother had not received the SARS-CoV2 vaccine during pregnancy. No abnormalities were observed during the perinatal period. On day 44 after birth, her sister, brother, mother, and aunt all tested positive for SARS-CoV2. On day 47, the patient developed fever with positive SARS-CoV2 antigen quantification. On admission, she showed good humor, with a body temperature of 38.4 °C and SpO2 of 100% in room air. The patient did not show any difficulty in breathing or feeding. Blood tests showed elevated AST (31 IU/L), LDH (254 IU/L), D-dimer (1.1 ng/mL), and ferritin (356 ng/mL) levels (Table 1). She defervesced on the third day of admission and was in good general condition. On day 5, blood tests showed AST (30 IU/L), LDH (292 IU/L), D-dimer (1.2 ng/mL), and elevated ferritin levels of 765 ng/mL. She defervesced on the 6th day of admission, with good general condition. On day 7, blood tests showed elevated AST (33 IU/L), LDH (289 IU/L), D-dimer (0.9 ng/mL), and ferritin (602 ng/mL), and she was discharged from the hospital. On day 36, she showed AST (38 IU/L), LDH (272 IU/L), D-dimer (0.7 ng/mL), and ferritin (256 ng/mL) (Table 1).

Case 4: A fifty-four-day-old girl was admitted to the hospital due to SARS-CoV-2 infection with pale stool. She was born at 41 weeks of gestation with a birth weight of 2830 g. Her mother had received the SARS-CoV2 vaccine twice during late pregnancy. No abnormalities were noted during the perinatal period or at 1-month follow-up. Thirty-six days after birth, her mother tested positive for SARS-CoV2. The mother and daughter were separated at home and the infant was changed from breastfeeding to formula. On the first day, she had a positive SARS-CoV2 PCR test result, and her stool turned gray. On admission, the patient showed good humor without any difficulty breathing or feeding. Blood tests showed normal bile duct system and liver enzymes, but elevated serum ferritin level as follows: total bilirubin (0.6 mg/dL), gamma-glutamyl transpeptidase (63 IU/L), AST (73 IU/L), ALT (68 IU/L), LDH (254 IU/L), and ferritin (398 ng/mL) (Table 1). She had a cough and was in good general condition. On day 4, no worsening of laboratory data was observed, and her stool returned to normal. On day 5, the patient was discharged, and on day 12, the blood test results at the outpatient clinic showed improvement (Table 1).

## 3. Discussion

In Japan, the COVID-19 outbreak began in February 2020, subsequently causing approximately nine million infections and 31,000 deaths until June 2022. At the beginning of the outbreak in Japan, of the hospitalized cases, 1038 pediatric patients with COVID-19 did not require positive pressure respiratory management, and only 2.1% required oxygen administration. In addition, 17/1038 (1.6%) of patients were less than 3 months old [10]. However, after the outbreak of the Omicron strain, which is more infectious than previous strains, the number of pediatric cases has been increasing [11]. Children under 5 years of age have been reported to generally exhibit mild symptoms [12], while newborns show more severe symptoms than infants [13]. Since our institution is the only pediatric inpatient facility in the region, most children with SARS-CoV-2 infections who require hospitalization present to our hospital. The hospital admission criteria included underlying medical conditions, disease onset at less than three months of age, and other cases with difficulty in home care. In our region, the Omicron strain was identified for the first time in January 2021, and the number of pediatric cases requiring hospitalization has been increasing in the last few months [11]. Although none of the cases in this study underwent genomic analysis to identify the causative strain, they all occurred during the pandemic caused by the Omicron strain.

In our cases, as shown in Table 1, the patients were mostly in good condition with fever, upper respiratory symptoms such as nasal discharge and cough, gastroenteritis, or poor feeding, without evidence of progressive hypoxemia or apneic attacks. None of the patients showed symptom aggravation after hospitalization because we tolerated “overtriage” of the symptoms and lab findings at that time when we had not experienced neonatal COVID-19 cases previously in Japan. In contrast, hyperferritinemia was observed in all four cases, which represents a higher frequency than that previously reported in patients under 21 years of age [14]. Serum ferritin levels peaked on the fifth to the seventh day from disease onset, without concurrent elevations in WBC, CRP, AST, ALT, or LDH. Regardless of ferritin levels, symptoms improved within 2–4 days of onset in most cases. Additionally, elevated transaminases were observed in two out of four cases. In both cases, the initial blood test values were the highest and improved over time.

It has been reported that SARS-CoV-2 exists in the monocytes of COVID-19 patients [15]. Since serum ferritin is secreted by macrophages following induction by inflammatory cytokines, its level could be a more sensitive marker for acute inflammatory responses in COVID-19 than other viral infections. In addition, ferritin itself has been suggested to serve as a pro-inflammatory cytokine that activates nuclear factor-kappa B [16]. Thus, ferritin induced by inflammatory cytokines may further induce the production of inflammatory cytokines [17]. Additionally, an increase in ferritin levels may be the result of the desaturation in part in these children [18].

MIS-C and MIS-N are multisystem diseases associated with post-coronavirus infection. The Centers for Disease Control and Prevention (CDC) diagnostic criteria for MIS-C are as follows: recent or current suspected COVID-19 in a child under 21 years of age with fever, two or more organ dysfunctions, and elevated inflammatory markers on blood tests [19]. There were 6851 cases of MIS-C reported to the CDC, of which 59 (0.86%) resulted in fatality [20]. Hyperferritinemia is a feature of both MIS-C and MIS-N [21]. Hyperferritinemia induced by COVID-19, as in our report, may trigger the production of inflammatory cytokines, thus triggering MIS-C and MIS-N. In our report, serum ferritin levels were elevated without other signs of hypercytokinemia, and no other MIS-C/MIS-N-like symptoms were observed. Additionally, the elevated levels of ferritin were very mild in our cases compared to the previous reports [22,23]. Only a few cases of MIS-C/MIS-N have been reported in Asia [20]. Genetic differences may contribute to the occurrence/severity of the disease.

The limitation of our study is that we had a small sample size to analyze. Further, our admission criteria were based on the “overtriage” of the symptoms and the lab findings because of the specific conditions of the COVID-19 era. Several reports have shown an association with apneic attack in early infants with SARS-CoV2 [24,25,26]; so, it is necessary to carefully monitor their clinical course and monitor the patients.

## 4. Conclusions

We carefully observed the four patients with the mild symptoms with mild elevation of ferritin or other inflammatory markers and found that they might usually show a stable condition if their lab data did not show a progressive exacerbation within 1 week after onset. If hyperferritinemia is observed in infantile COVID-19 patients, their clinical course should be monitored closely.

## Figures and Tables

**Table 1 children-10-00874-t001:** Patients’ characteristics and laboratory data with clinical courses.

			Case 1	Case 2	Case 3	Case 4
Patients	Age		38	54	47	28
	Weight		4488	4258	4748	4576
	Gender		f	f	f	f
Perinatal information	Gestational age (weeks)		41	39	37	41
	Birth weight (g)		3348	2250	2600	2830
	Vaccination during pregnancy		no	no	no	twice
	Feeding		mixed	infant formula	infant formula	infant formula
Symptoms	Fever		-	1 day	1 day	-
	Cough		+	+	-	+
	Nasal discharge		+	-	-	+
	Poor feeding		-	±	-	-
	Other		-	-	-	White stool
Radiological findings			-	-	-	-

Laboratory data on admission		Standards	day 4	day 1	day 1	day 1
	CRP (mg/L)	0–0.14	0.06	0.17	0.05	0.02
	Ferritin (ng/mL)	25–280 (boy) 10–120 (girl)	570	280	356	398
	AST (U/L)	22–66	146	29	31	73
	ALT (U/L)	13–56	87	15	32	68
	LDH (U/L)	205–418	370	253	254	254
	D-dimer (μg/mL)	0.11–0.42	N/A	1	1.1	0.9
	Fibrinogen (g/L)	82–383	N/A	2.38	3.43	2.06
	WBC (/μL)	4600–18,900	8400	7800	3900	6200
	Hb (g/dL)	9.5–13.7	15	10	9.6	11.5
	PLT (10^4^/μL)	25–82	36	35.9	45	41.7
	TP (g/dL)	5.1–6.8	6.1	5.4	5.7	5.4
	Alb (g/dL)	3.1–4.6	3/7	3.7	3.9	3.7
	T-Bil (mg/dL)	0.1–0.8	0.6	0.6	0/4	0.6
	γ-GTP (U/L)	15–150	139	N/A	61	63
	BUN (mg/dL)	2.2–14.1	8	8	9	12
	CRE (mg/dL)	0.12–0.27	0.22	0.18	0.24	0.23
	Na (mmol/L)	135–145	137	136	140	137
	K (mmol/L)	4.1–5.6	5.1	4.8	5.4	4.8
	Cl (mmol/L)	101–111	103	102	108	105
	CPK (U/L)	43–321	67	93	131	N/A
	RBC (μL)	340–500	458	313	307	355
	Ht (%)	28.5–41.1	44.1	29.6	27.9	33.7
	PT (%)	115–153	N/A	>130	>130	114
	APTT (sec)	35.1–46.3	N/A	38.3	<20	32.5
	FDP (μg/mL)	0–4.9	N/A	2.6	3.8	2.9
		Standards	day 10	day 7	day 5	day 5
Laboratory data during the clinical course	CRP (mg/dL)	0–0.14	0.48	1.1	<0.01	0.02
	Ferritin (ng/mL)	25–280 (boy) 10–120 (girl)	368	332	765	365
	AST (U/L)	22–66	91	42	30	41
	ALT (U/L)	13–56	82	34	29	58
	LDH (U/L)	205–418	300	359	292	236
	D-dimer	0.11–0.42	0.9	1.2	1.2	0.8
	Fib	82–383	300	398	284	189
	WBC (/μL)	4600–18,900	7600	9500	7000	7000
	Hb (g/dL)	9.5–13.7	13.3	10	10.3	11.9
	PLT (10^4^/μL)	25–82	48.8	13.5	33.9	55.4
	TP (g/dL)	5.1–6.8	5.7	5.8	5.6	5.6
	Alb (g/dL)	3.1–4.6	3.5	3.5	3.9	3.8
	T-Bil (mg/dL)	0.1–0.8	0.9	0.3	0.3	0.7
	γ-GTP (U/L)	15–150	121	26	70	65
	BUN (mg/dL)	2.2–14.1	11	4	8	10
	CRE (mg/dL)	0.12–0.27	0.37	0.15	0.24	0.25
	Na (mmol/L)	135–145	139	140	139	136
	K (mmol/L)	4.1–5.6	5.3	5.7	5.5	5.4
	Cl (mmol/L)	101–111	106	107	107	104
	CPK (U/L)	43–321	67	61	141	N/A
	RBC (/μL)	340–500	458	313	316	371
	Ht (%)	28.5–41.1	39.3	29.6	30.3	35.2
	PT (%)	115–153	118	111	121	>130
	APTT (sec)	35.1–46.3	35.1	33	35.9	26.8
	FDP (μg/mL)	0–4.9	2.7	3.6	3.0	2.7

## Data Availability

The data that support the findings of this study are available from the corresponding author, [C.O.], upon reasonable request.

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
