# Peer review of "Incidental Hyperferritinemia in Very Young Infants with Mild Symptoms of COVID-19 Disease"

_children, 2023, doi:10.3390/children10050874_

Round 1
Reviewer 1 Report
The basal problem of the article is the lack of laboratory standards, even for the study. Reference values ​​are not given even for the study that is the basis of this article. It is therefore difficult to relate to the significance of the observations. In some cases, the increase in ferritin appears to be borderline or very mild. The reviewer refers them to commonly known references. It is possible that the hospital where the authors work has different standards.
It is therefore difficult to ascertain the significance of the publication. Ferritin is also an inflammatory protein, and the children referred to the ward showed signs of infection.
Line 34 references needed. Sugested:
J Clin Med. 2021 Oct 30;10(21):5098. doi: 10.3390/jcm10215098.
Sci Rep. 2022 May 11;12(1):7765. doi: 10.1038/s41598-022-11068-0.
line 37 Description applies to America and Europe, no European references. Suggested:
Pediatric rheumatology online journal vol. 19,1 29. 16 Mar. 2021, doi:10.1186/s12969-021-00511-7
The Lancet. Child & adolescent health vol. 4,9 (2020): 669-677. doi:10.1016/S2352-4642(20)30215-7
The Lancet regional health. Europe vol. 19 (2022): 100443. doi:10.1016/j.lanepe.2022.100443
line 136 : Very strange word devision
line 142: The sentence meaning is difficult to understand.
lines 193-6 Were the levels of anti-Sars CoV 2 antibodies measured in children of vaccinated mothers? The observation made in 3 cases is not very reliable (especially without concentration measurement).
line 207 Less transparent conclusions. What do the authors want to emphasize? Simple increase in ferritin in 4 patients?
The article is an observation of 6 patients. It is difficult to classify whether it should be a case report or a review. In a review situation, the study group is very small. It is worth pointing out the limitations of the analysis related to this small group of patients.
Author Response
R1: The basal problem of the article is the lack of laboratory standards, even for the study. Reference values ​​are not given even for the study that is the basis of this article. It is therefore difficult to relate to the significance of the observations. In some cases, the increase in ferritin appears to be borderline or very mild. The reviewer refers them to commonly known references. It is possible that the hospital where the authors work has different standards.
It is therefore difficult to ascertain the significance of the publication. Ferritin is also an inflammatory protein, and the children referred to the ward showed signs of infection.
A1: Thank you very much for your precious comments. We added the reference values of the labo data in the Table. Also, we added the criteria which we decided at that point for patients’ admission.
R2: Line 34 references needed. Sugested:
J Clin Med. 2021 Oct 30;10(21):5098. doi: 10.3390/jcm10215098.
Sci Rep. 2022 May 11;12(1):7765. doi: 10.1038/s41598-022-11068-0.
A2: Thank you for your suggestion. We added the references as 2 and 3.
R3: Line 37 Description applies to America and Europe, no European references. Suggested:
Pediatric rheumatology online journal vol. 19,1 29. 16 Mar. 2021, doi:10.1186/s12969-021-00511-7
The Lancet. Child & adolescent health vol. 4,9 (2020): 669-677. doi:10.1016/S2352-4642(20)30215-7
The Lancet regional health. Europe vol. 19 (2022): 100443. doi:10.1016/j.lanepe.2022.100443
A3: Thank you for your suggestion. We added the references as 5, 6, and 7.
R4: line 136 : Very strange word devision
A4: Thank you for your point. We changed the sentence Marked as red.
R5: line 142: The sentence meaning is difficult to understand.
A5: Thank you for your point. We changed the sentence Marked as red.
R6: lines 193-6 Were the levels of anti-Sars CoV 2 antibodies measured in children of vaccinated mothers? The observation made in 3 cases is not very reliable (especially without concentration measurement).
A6: We appreciate your indication. We did not measure the concentration of the antibodies. We remove the speculation between maternal vaccination and ferritin levels. We also removed Figure 1.
R7: line 207 Less transparent conclusions. What do the authors want to emphasize? Simple increase in ferritin in 4 patients?
A7: We changed the point to emphasize in the conclusion. We experienced the severe case with the elevated serum ferritin level and progressive pneumonia in a newborn case. Also, there are reports of apneic attacks in neonatal cases. Thus we carefully observed the 6 patients with the mild symptoms with mild elevation of ferritin or other inflammatory markers and found that they might show stable condition if their labo data did not show the exacerbation.
R8: The article is an observation of 6 patients. It is difficult to classify whether it should be a case report or a review. In a review situation, the study group is very small. It is worth pointing out the limitations of the analysis related to this small group of patients.
A8: Thank you very much for your comments. We added the limitation in the discussion section.

Reviewer 2 Report
Dear colleagues, according to WHO data Omicron variant of concern was designated as such by Technical Advisory Group on Virus Evolution (TAG-VE) on 26th November 2021. - www.who.int/news/item/28-11-2021-update-on-omicron
You are stating in row 44 that you included patients in your study during mid-2021 till the beginning of 2022. None of the presented patients has a more precise temporal location of disease.
So your statement in rows 160-162 ["Although none of the cases in this study underwent genomic analysis to identify the causative strain, they all occurred during the pandemic caused by the Omicron strain"] can not be substantiated.
Your claim in rows 158-160 ["the Omicron strain was identified for the first time in January 2021, and the number of paediatric cases requiring hospitalization has been increasing in the last few months (6)"] in erroneous in terms of timing. It was not possible to have Omicron VOC circulation starting January 2021 in your region as TAG-VE named this variant with this name almost a year later.
Please correct data and rephrase section in rows 147-160. Here is mandatory to provide more accurate time line for paediatric cases presented in reference 5, because these children were admitted in previous waves, unrelated with Omicron circulation, since this paper was published in 2021.
How many children were admitted in your country/county in Omicron wave?
Regarding cases presented I am confused about the reason for managing as in-patients these children
Case 1 - Due to the increasing nasal discharge, the patient could not sleep well.... She had a body temperature of 36.5°C, SpO2 of 97% on room air, and heart rate of 130-140 bpm without any respiratory distress. No crackles or wheezing were heard on auscultation. Chest radiography revealed no sign of pneumonia, bronchiolitis, or heart failure.
Normal vital signs, no fever, no distress and normal [I would say with questionable indication?!] chest radiographic findings. Why was in hospital for 5 days this child and why she had numerous lab investigations ?
Case 2 - a small-for-gestational age child [2250 grams birth weight at 39 weeks gestational age] with a more important clinical involvement than case 1 but "On physical examination, she did not show any symptoms of respiratory distress, with an SpO2 of 98%." In spite of having some mild symptomatology you are stating an rapid and excellent outcome "Blood tests, including inflammatory markers and urinalysis, showed no abnormal findings. On the second day of hospitalization (the second day of onset), her fever resolved and her general condition improved". Why was she kept in hospital another 10 days? What is the current protocol for COVID-19 paediatric patients management in the absence of respiratory involvement? Do you routinely perform chest CT in these patients? In what age group?
Case 3 - a slightly premature child [37 weeks gestational age] presented with "... good humor, with a body temperature of 38.4 °C and SpO2 of 100% in room air. The patient did not show any difficulty in breathing or feeding." Why was she admitted and kept in hospital for 7 days?
Case 4 - child older than 90 days [even if we use the fever without origin age limits for being more cautious this child is marginally older] and presented with "good humor with a body temperature of 38.1℃ and SpO2 of 99% at room air. Although her fever lasted for two days, she did not show any difficulty breathing or feeding. Blood tests revealed no abnormalities". Why was she admitted and kept in hospital for 6 days?
Case 5 - had a presenting feature "pale stools" that could explain a certain level of concern, but "patient showed good humor without any difficulty breathing or feeding. Blood tests showed normal bile duct system and liver enzymes..." In spite of these laboratory findings and of good clinical status child remained in hospital for 5 days.
Case 6 - seems the only case in the present series of cases that had a potentially serious condition, prior to transfer, but had a very rapid improvement and normal ferritin, in spite of being the case with the most important clinical inflammation.
May I ask why these children had ferritin tested in spite of being oligosymptomatic? What is the cost-efficacy of such an approach? What potential serious scenarios are prevented in mild paediatric COVID-19 cases by routinely performing inflammatory markers in spite of rapid clinical improvement?
Did you had retested any of these discordant ferritin values by other methods or in another lab? Do you totally rule-out an "innocent" lab finding?
Figure 1 has to be redrawn because it could be misleading. On the horizontal axis you have placed twice time from onset...
Conclusion stated in submitted paper "There is a high possibility that hyperferritinemia may occur in young patients infected with SARS-CoV2 infection despite their general condition, and it may thus be necessary to carefully monitor their progress and 208 follow the patient's ferritin levels until they normalize" is speculative and not validated by presented data - to small sample size. Literature data available do not support such a claim of immune response in very young children.
In a paper [Bourgeois FT, Gutiérrez-Sacristán A, Keller MS, Liu M, Hong C, Bonzel CL, Tan ALM, Aronow BJ, Boeker M, Booth J, Cruz Rojo J, Devkota B, García Barrio N, Gehlenborg N, Geva A, Hanauer DA, Hutch MR, Issitt RW, Klann JG, Luo Y, Mandl KD, Mao C, Moal B, Moshal KL, Murphy SN, Neuraz A, Ngiam KY, Omenn GS, Patel LP, Jiménez MP, Sebire NJ, Balazote PS, Serret-Larmande A, South AM, Spiridou A, Taylor DM, Tippmann P, Visweswaran S, Weber GM, Kohane IS, Cai T, Avillach P; Consortium for Clinical Characterization of COVID-19 by EHR (4CE). International Analysis of Electronic Health Records of Children and Youth Hospitalized With COVID-19 Infection in 6 Countries. JAMA Netw Open. 2021 Jun 1;4(6):e2112596. doi: 10.1001/jamanetworkopen.2021.12596. Erratum in: JAMA Netw Open. 2021 Jul 1;4(7):e2122388. PMID: 34115127; PMCID: PMC8196345.] presenting a cohort of 671 patients with more than 27.000 laboratory values available in electronic databases from 6 countries, ferritin average level was 417 ng/mL so only one of your patients has an seriously elevated ferritin level.
Author Response
R1: Dear colleagues, according to WHO data Omicron variant of concern was designated as such by Technical Advisory Group on Virus Evolution (TAG-VE) on 26th November 2021. - www.who.int/news/item/28-11-2021-update-on-omicron
You are stating in row 44 that you included patients in your study during mid-2021 till the beginning of 2022. None of the presented patients has a more precise temporal location of disease.
So your statement in rows 160-162 ["Although none of the cases in this study underwent genomic analysis to identify the causative strain, they all occurred during the pandemic caused by the Omicron strain"] can not be substantiated.
Your claim in rows 158-160 ["the Omicron strain was identified for the first time in January 2021, and the number of paediatric cases requiring hospitalization has been increasing in the last few months (6)"] in erroneous in terms of timing. It was not possible to have Omicron VOC circulation starting January 2021 in your region as TAG-VE named this variant with this name almost a year later.
Please correct data and rephrase section in rows 147-160. Here is mandatory to provide more accurate time line for paediatric cases presented in reference 5, because these children were admitted in previous waves, unrelated with Omicron circulation, since this paper was published in 2021.
How many children were admitted in your country/county in Omicron wave?
A1: Thank you for your detailed points. We examined the data of the patients and found that the patients were admitted from January to May, 2022. We made mistake because we had the very first case (not including here) in August 2021. We changed the date in the introduction and method sections.
R2: Regarding cases presented I am confused about the reason for managing as in-patients these children
Case 1 - Due to the increasing nasal discharge, the patient could not sleep well.... She had a body temperature of 36.5°C, SpO2 of 97% on room air, and heart rate of 130-140 bpm without any respiratory distress. No crackles or wheezing were heard on auscultation. Chest radiography revealed no sign of pneumonia, bronchiolitis, or heart failure.
Normal vital signs, no fever, no distress and normal [I would say with questionable indication?!] chest radiographic findings. Why was in hospital for 5 days this child and why she had numerous lab investigations ?
A2: I totally agree with your points. Before this first case, we experienced the rapid exacerbation case of the neonate after the mild symptoms and lab data. Thus we would like to check the changes of the symptoms and the progressive exacerbation of the lab data. In addition, because of the several reports of apneic cases of the neonatal COVID-19 cases, we observed the patients at least within the first 1 week in the hospital.
R3: Case 2 - a small-for-gestational age child [2250 grams birth weight at 39 weeks gestational age] with a more important clinical involvement than case 1 but "On physical examination, she did not show any symptoms of respiratory distress, with an SpO2 of 98%." In spite of having some mild symptomatology you are stating an rapid and excellent outcome "Blood tests, including inflammatory markers and urinalysis, showed no abnormal findings. On the second day of hospitalization (the second day of onset), her fever resolved and her general condition improved". Why was she kept in hospital another 10 days? What is the current protocol for COVID-19 paediatric patients management in the absence of respiratory involvement? Do you routinely perform chest CT in these patients? In what age group?
A3: Basically the same reason as A2, we observed the patient for 10 days in the hospital. In this case, in addition, the patient showed a slight exacerbation of the lab data. We were concerned about the former severe case aforementioned at that point when we had not experienced many of such cases. Now we recognized that the CT scanning for this case was overtriaged as you suggested.
R4: Case 3 - a slightly premature child [37 weeks gestational age] presented with "... good humor, with a body temperature of 38.4 °C and SpO2 of 100% in room air. The patient did not show any difficulty in breathing or feeding." Why was she admitted and kept in hospital for 7 days?
A4: Thank you for your comments. We observed the patients in the hospital within a week with the same reason as aforementioned and also with concern of apnea because of prematurity. We added the definitions and criteria of the admission in the method section.
R5: Case 4 - child older than 90 days [even if we use the fever without origin age limits for being more cautious this child is marginally older] and presented with "good humor with a body temperature of 38.1℃ and SpO2 of 99% at room air. Although her fever lasted for two days, she did not show any difficulty breathing or feeding. Blood tests revealed no abnormalities". Why was she admitted and kept in hospital for 6 days?
A5: As you suggested, to observe this patient for a week is way too long in the usual situation. We would like to observe the patient 2-3 days after decline of the fever because sometimes the upper respiratory symptoms exacerbate in this period with increasing sputum.
R6: Case 5 - had a presenting feature "pale stools" that could explain a certain level of concern, but "patient showed good humor without any difficulty breathing or feeding. Blood tests showed normal bile duct system and liver enzymes..." In spite of these laboratory findings and of good clinical status child remained in hospital for 5 days.
A6: Thank you for your comments. We observed the patient for basically the aforementioned reasons. Further, since the “pale stool” is the first episode of these COVID-19 patient, we observed carefully for 5 days.
R7: Case 6 - seems the only case in the present series of cases that had a potentially serious condition, prior to transfer, but had a very rapid improvement and normal ferritin, in spite of being the case with the most important clinical inflammation.
May I ask why these children had ferritin tested in spite of being oligosymptomatic? What is the cost-efficacy of such an approach? What potential serious scenarios are prevented in mild paediatric COVID-19 cases by routinely performing inflammatory markers in spite of rapid clinical improvement?
A7: Thank you for your precise comments. In these periods, we accumulated the neonatal COVID-19 cases and basically allowed “overestimation” because we did not know the “standard” clinical course of these patients in Japan. Our previous severe case cost more with PICU management. We believe that if we could find the exacerbation point within the first 1 week in the hospital, final “cost” is much more inexpensive in the Japanese insurance system.
R8: Did you had retested any of these discordant ferritin values by other methods or in another lab? Do you totally rule-out an "innocent" lab finding?
A8: We added the retested data in the Table. We did not rule-out “innocent” lab finding but improving lab data.
R9: Figure 1 has to be redrawn because it could be misleading. On the horizontal axis you have placed twice time from onset…
A9: Thank you for your suggestion. We removed the figure.
R10: Conclusion stated in submitted paper "There is a high possibility that hyperferritinemia may occur in young patients infected with SARS-CoV2 infection despite their general condition, and it may thus be necessary to carefully monitor their progress and follow the patient's ferritin levels until they normalize" is speculative and not validated by presented data - to small sample size. Literature data available do not support such a claim of immune response in very young children.
A10: Thank you for your suggestion. We changed the conclusion and added the papers suggested (ref 22, 23).

Round 2
Reviewer 1 Report
Please note the double numbering of references. It needs improvement.
Author Response
R1: Please note the double numbering of references. It needs improvement.
A1: Thank you for your comment. We corrected the numbering.
Reviewer 2 Report
Dear colleagues thank you for answers and corrections.
Some issues are still pending.
In table inserted at row 83 you are listing the 6 infants [one new-born patient, case 5 and 5 older patients] so it would be more appropriate to rephrase title by deleting "neonatal" ... Hyperferritinemia in neonatal/infantile COVID-19 patients with mild symptoms
Maybe "Incidental hyperferritinemia in very young infants with mild symptoms of COVID-19 disease" would be a reasonable alternative... Or any similar rephrasing.
You mentioned reason for admitting neonatal/very young infant cases in January to May 2022 by referencing a 14 days old infant with apnea. This SARS-CoV-2 and Influenza B coinfection case was published in 15 March 2022 [Maddali R, Cervellione KL, Lew LQ. Apnea in a Two-Week-Old Infant Infected with SARS-CoV-2 and Influenza B. Case Rep Pediatr. 2022 Mar 15;2022:2969561. doi: 10.1155/2022/2969561. PMID: 35340538; PMCID: PMC8941565.], so please make needed correction.
Second reference used [reference 11] was published just one month ago [Paolin C, Zanetto L, Frison S, Boscolo Mela F, Tessari A, Amigoni A, Daverio M, Bonardi CM. Apneas requiring respiratory support in young infants with COVID-19: a case series and literature review. Eur J Pediatr. 2023 Mar 13:1–6. doi: 10.1007/s00431-023-04856-x. Epub ahead of print. PMID: 36912961; PMCID: PMC10009862.] so it was not possible to be of your awareness in 2022 that authors will publish it one year later such a paper. Please rephrase entirely this section and insert these references eventually in discussion section not in Methods section because methodology of a certain paper should be part of a strategic plan generated before research is done.
Both references [10 and 11] are presented children as old as 2 months corrected-age so case 4, that has normal ferritin levels and more than 3 months of age, should be deleted from this paper because it is not respecting presumed inclusion criteria [age and ferritin] but had only trivial symptoms (actually a case for a General practitioner monitoring not for hospital admittance...).
Case 6 does not fit also inclusion criteria, excepting minimal disease severity. Please clarify why did you include in your analysis an infant with normal ferritin levels?
To be more specific: you are presenting 6 infants with mild disease but only 4 of them had a documented significant increase in ferritin levels (as per values in table and note in rows 179-180). In table values listed for cases 4 and 6 are normal if values are presented in ng/mL according to lab standards inserted by you. Please clarify Laboratory standards posted because there are two distinct sets of values and only one type of units allocated to these sets of values.
In rows 186-192 you are presenting potential mechanisms that could explain increase of ferritin levels in these children. You can add hypoxemia a another mechanism that has been proven in animal models [Larede and Storey paper cited above] as factor for increasing ferritin, thus explaining these findings, at least in these children - at least in case 1 that has the highest ferritin level and mild desaturation [row 70]
Larade K, Storey KB. Accumulation and translation of ferritin heavy chain transcripts following anoxia exposure in a marine invertebrate. J Exp Biol. 2004 Mar;207(Pt 8):1353-60. doi: 10.1242/jeb.00872. PMID: 15010486.
Summary should also be changed because you documented 4 not 6 infants with increase in ferritin levels.
reference section should be renumbered because it has a significant numbering issue [double numbers].
Author Response
R1: In table inserted at row 83 you are listing the 6 infants [one new-born patient, case 5 and 5 older patients] so it would be more appropriate to rephrase title by deleting "neonatal" ... Hyperferritinemia in neonatal/infantile COVID-19 patients with mild symptoms
Maybe "Incidental hyperferritinemia in very young infants with mild symptoms of COVID-19 disease" would be a reasonable alternative... Or any similar rephrasing.
A1: Thank you for your precious suggestion. We changed the title as suggested.
R2: You mentioned reason for admitting neonatal/very young infant cases in January to May 2022 by referencing a 14 days old infant with apnea. This SARS-CoV-2 and Influenza B coinfection case was published in 15 March 2022 [Maddali R, Cervellione KL, Lew LQ. Apnea in a Two-Week-Old Infant Infected with SARS-CoV-2 and Influenza B. Case Rep Pediatr. 2022 Mar 15;2022:2969561. doi: 10.1155/2022/2969561. PMID: 35340538; PMCID: PMC8941565.], so please make needed correction.
R3: Second reference used [reference 11] was published just one month ago [Paolin C, Zanetto L, Frison S, Boscolo Mela F, Tessari A, Amigoni A, Daverio M, Bonardi CM. Apneas requiring respiratory support in young infants with COVID-19: a case series and literature review. Eur J Pediatr. 2023 Mar 13:1–6. doi: 10.1007/s00431-023-04856-x. Epub ahead of print. PMID: 36912961; PMCID: PMC10009862.] so it was not possible to be of your awareness in 2022 that authors will publish it one year later such a paper. Please rephrase entirely this section and insert these references eventually in discussion section not in Methods section because methodology of a certain paper should be part of a strategic plan generated before research is done.
R4: Both references [10 and 11] are presented children as old as 2 months corrected-age so case 4, that has normal ferritin levels and more than 3 months of age, should be deleted from this paper because it is not respecting presumed inclusion criteria [age and ferritin] but had only trivial symptoms (actually a case for a General practitioner monitoring not for hospital admittance...).
A2/A3/A4: We appreciate your comments. We removed the reference 10, 11 and changed into 10. Loron G, Tromeur T, Venot P, Beck J, Andreoletti L, Mauran P, et al. COVID-19 Associated With Life-Threatening Apnea in an Infant Born Preterm: A Case Report. Front Pediatr. 2020 Sep 15;8:568. We added original references 10 and 11 into discussion section as references 24 and 25.
R5: Case 6 does not fit also inclusion criteria, excepting minimal disease severity. Please clarify why did you include in your analysis an infant with normal ferritin levels?
To be more specific: you are presenting 6 infants with mild disease but only 4 of them had a documented significant increase in ferritin levels (as per values in table and note in rows 179-180). In table values listed for cases 4 and 6 are normal if values are presented in ng/mL according to lab standards inserted by you. Please clarify Laboratory standards posted because there are two distinct sets of values and only one type of units allocated to these sets of values.
A5: Thank you for your comments. We totally agree with your comments. We removed the case 4 and 6 with normal ferritin levels. Also, 2 values of ferritin standards show that of boys and girls. Thus we changed the table.
R6: In rows 186-192 you are presenting potential mechanisms that could explain increase of ferritin levels in these children. You can add hypoxemia a another mechanism that has been proven in animal models [Larede and Storey paper cited above] as factor for increasing ferritin, thus explaining these findings, at least in these children - at least in case 1 that has the highest ferritin level and mild desaturation [row 70]
Larade K, Storey KB. Accumulation and translation of ferritin heavy chain transcripts following anoxia exposure in a marine invertebrate. J Exp Biol. 2004 Mar;207(Pt 8):1353-60. doi: 10.1242/jeb.00872. PMID: 15010486.
A6: Thank you for your suggestion. We added the suggested reference as ref. 18.
R7: Summary should also be changed because you documented 4 not 6 infants with increase in ferritin levels.
reference section should be renumbered because it has a significant numbering issue [double numbers].
A7: Thank you for your comment. We renumbered the references.